# On Reliability of a Double Redundant Renewable System with a Generally Distributed Life and Repair Times

**Vladimir Rykov [1,2]**, **Dmitry Efrosinin [2,3,]**, **Natalia Stepanova [4]** and **Janos Sztrik [5]**

[1] Department of Applied Mathematics and Computer Modelling, Faculty of Automation and Computer Science, Gubkin Russian State University of Oil and Gas, 119991 Moscow, Russia; vladimir_rykov@mail.ru

[2] Department of Information Technologies, Faculty of Mathematics and Natural Sciences, Peoples' Friendship University of Russia (RUDN University), 117198 Moscow, Russia

[3] Insitute for Stochastics, Johannes Kepler University Linz, 4030 Linz, Austria

[4] Laboratory N17, Trapeznikov Institute of Control Sciences of RAS, 117997 Moscow, Russia; natalia0410@rambler.ru

[5] Department of Informatics and Networks, Faculty of Informatics, University of Debrecen, 4032 Debrecen, Hungary; sztrik.janos@inf.unideb.hu

* Correspondence: dmitry.efrosinin@jku.at

**Abstract:** The paper provides reliability analysis of a cold double redundant renewable system assuming that both life-time and repair time distributions are arbitrary. The proposed approach is based on the theory of decomposable semi-regenerative processes. We derive the Laplace–Stieltjes transform of two main reliability measures like the distribution of the time between failures and the time to the first failure. The transforms are used to calculate corresponding mean times. It is further derived in closed form the time-dependent and time stationary state probabilities in terms of the Laplace transforms. Numerical results illustrate the effect of the type of distributions as well as their parameters on the derived reliability and probabilistic measures.

**Keywords:** redundant system; reliability; arbitrary distributions; time-dependent characteristics; decomposable semi-regenerative process

## 1. Introduction

The use of parallel or redundant units is an often recommended way of increasing the system reliability. In the last few decades, many authors have studied the redundant systems under different sets of assumptions about the system's attributes and level of generality, the distribution of life, and repair times when one or more units are standby. In this paper, we formulate the model where both of distributions are assumed to be arbitrary. Such models are interesting both from theoretical and practical points of view. Theoretically, these studies are related to the creation and development of new mathematical methods such as regenerative methods, especially methods of the Decomposable Semi-Regenerative Processes (DSRP), and Markovization methods via introduction of the supplementary variables. In practical terms, the tasks under consideration form a basis for analyzing the reliability of different complex systems with generally distributed times.

Smith's idea about regenerative stochastic processes [1] found many generalizations and applications. This approach still helps to solve many applied problems since it reduces the complexity of the task. Instead of a whole time period, the process can be analyzed within a separate regenerative period of time. However, if the process describing the system behavior is complicated enough in order to be exhaustively analyzed on the regenerative period, even in this case, different generalizations



of the regenerative idea are possible. One possible generalization is to combine Smith's idea with a semi-Markov approach [2], which seems quite appropriate for real models. The result of such combination is a semi-regenerative process. In Rykov and Yastrbrnrtsky [3], it appears under the name "Regenerative processes with several types of regeneration points". Klimov [4] considered it as a "semi-Markov process with additional trajectories". In accordance with Jakod [5] and Nummeline [6], in this paper, we will call such process a semi-regenerative process (SRP).

The next possible generalization is to construct within the basic regeneration cycle other regeneration points called embedded regeneration points. Sometimes, in a particular model, one can find some embedded regeneration points and to represent the process distribution in a separate regeneration cycle in terms of its distribution in embedded regeneration cycles of the second level. Initially, such processes appeared in [7,8] under the name "Regenerative processes with embedded regeneration cycles" or "Decomposable Semi-Regenerative Processes" (DSRP). Regenerative processes with embedded regeneration cycles are widely used applications in operations research. In [7], regenerative processes have been applied to study priority queueing systems. In [9,10], the one-server queueing system with infinite queue, recurrent input flow, and general service time distribution ($GI|GI|1, \infty$—system in Kendall's notation [11]) has been studied in terms of these processes (see also [12]). In paper [13], the authors investigated a model of a polling system by means of the DSRP. Another approach related to the study of complex systems (proposed in Belyaev [14]) contains the introduction of so-called supplementary variables. According to this method, the original model can be described by a continuous-time Markov chain (Markovization method). The implementation of this method to study reliability models is proposed by Rykov [15], Mazumdar [16], and Mine and Asakura [17].

The redundancy of some technical system means the duplication of critical components resulting in increasing reliability of the system. The double redundant systems are well known and were studied and applied under different assumptions like repair maintenance, preventive maintenance policy, exponential distributions for life and repair times, and so on; see, e.g., Gaver [18], Gnedenko et al. [19], Mine and Asakura [20], and Srinivasan [21]. In this paper, we study a homogeneous cold double redundant system with arbitrary life- and repair time distributions. In such a system, the cold standby redundancy means that the redundant component does not fail in a standby mode. The closed form solutions for the main reliability characteristics have been derived in terms of an integral transformation like a Laplace transform. The proposed analysis is based on the DSRP, the stochastic equations for the corresponding random variables (RVs), and the theory of regenerative processes [1].

The rest of the paper is organized as follows. In Section 2, the problem set, some assumptions, and notations are introduced. In Section 3, the Laplace–Stieltjes transforms (LSTs) of the regeneration period (the system life cycle), and the Laplace transforms (LTs) of the times to the first failure and between system failures are calculated. In Sections 4 and 5, the time-dependent system state probabilities (TDSSPs) and stationary system state probabilities (SSSPs) are calculated. Finally, using some numerical examples, we demonstrate the effect of type of distributions and their parameter on the reliability and probabilistic measures of the system in Section 6. The paper ends with a conclusion.

## 2. The Problem Set: Assumptions and Notations

Consider a homogeneous cold double redundant renewable system with generally distributed life- and repair times, which, according to modified Kendall's notations [11], will be denoted as $<GI_2/GI/1>$. The system consists of two identical units which can be in two possible states: operational and failed. The system fails when both of units are in a failed state. For the renewable system, different strategies of its renovation are possible. In this paper, we consider a strategy when a new system cycle begins at the point in time when one unit starts working while the other unit failed and is being repaired.

Denote by $A_i$ $(i = 1, 2, \dots)$, life times of the system units, by $B_i$ $(i = 1, 2, \dots)$, its repair times, and suppose that all these random variables (RVs) are mutually independent and identically distributed.

Thus, denote by $A(t) = \mathbb{P}[A_i \leq t]$ and $B(t) = \mathbb{P}[B_i \leq t]$ the corresponding cumulative distribution functions (CDFs). Denote also by $A$ and $B$ the RVs with the same CDFs as $A_i$ and $B_i$, respectively. Suppose that the instantaneous failures and repairs are impossible and their mean times are finite:

$$A(0) = B(0) = 0, \quad a = \int_0^\infty (1 - A(x))dx < \infty, \quad b = \int_0^\infty (1 - B(x))dx < \infty.$$

Denote by $E$ the set of system states. We assume that $E = \{i = 0, 1, 2\}$, where $i$ stands for the number of failed units, and we introduce a random process $J = \{J(t), t \geq 0\}$, where

$$J(t) = \text{number of failed units at time } t.$$

A simple state transition diagram is illustrated in Figure 1.

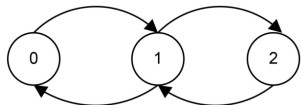

**Figure 1.** State transition diagram.

Denote by $F$ the time between system failures, i.e., the period of time the system begins to operate again up to the next system failure. We remind that the system is again in an operational state when operation of one unit and repair of another one begin simultaneously. Denote by $F_1$ the time to first system failure (the time from the epoch when both units are in operational state up to the system failure) (see Figure 2). The functions $F(t) = \mathbb{P}[F \leq t]$ and $F_1(t) = \mathbb{P}[F_1 \leq t]$ are corresponding CDFs.

We are interesting in calculation of the reliability function

$$R(t) = \mathbb{P}[F_1 > t] = 1 - F_1(t),$$

distribution of time to the first failure $F_1(t)$, time-dependent system state probabilities (TDSSPs)

$$\pi_j(t) = \mathbb{P}[J(t) = j], \quad j = 0, 1, 2,$$

and stationary system state probabilities (SSSPs)

$$\pi_j = \lim_{t \to \infty} \pi_j(t) \equiv \lim_{t \to \infty} \mathbb{P}[J(t) = j], \quad j = 0, 1, 2,$$

as well as the availability coefficient

$$K_{\text{av.}} = \pi_0 + \pi_1 = 1 - \pi_2.$$

The following notations will be used next.

- The Laplace–Stiltjes transforms (LSTs) of their CDFs are denoted as:

$$
\begin{aligned}
\tilde{a}(s) &= \mathbb{E}\left[e^{-sA}\right] = \int_0^\infty e^{-sx} dA(x), \quad \tilde{b}(s) = \mathbb{E}\left[e^{-sB}\right] = \int_0^\infty e^{-sx} dB(x), \\
\tilde{f}(s) &= \mathbb{E}\left[e^{-sF}\right] = \int_0^\infty e^{-sx} dF(x), \quad \tilde{f}_1(s) = \mathbb{E}\left[e^{-sF^{(1)}}\right] = \int_0^\infty e^{-sx} dF_1(x).
\end{aligned}
$$

- In the case when the repair completion comes before the failure of other unit and when an operative unit fails before the repair completion of other unit, the distributions are equal, respectively,

$$a_B(t) = \int_0^t B(x)dA(x) = \mathbb{P}[B \leq A \leq t], \quad b_A(t) = \int_0^t A(x)dB(x) = \mathbb{P}[A \leq B \leq t].$$

Applying the Laplace transform (LT) for $a_B(t)$ and $b_A(t)$, we get

$$\tilde{a}_B(s) = \int_0^\infty e^{-st}da_B(t) = \int_0^\infty e^{-sx}B(x)dA(x), \tag{1}$$

$$\tilde{b}_A(s) = \int_0^\infty e^{-st}db_A(t) = \int_0^\infty e^{-sx}A(x)dB(x).$$

- Their derivatives with sign minus in point zero are denoted by:

$$a_B = -\frac{d}{ds}\tilde{a}_B(s)\big|_{s=0} = \int_0^\infty xB(x)dA(x),$$

$$b_A = -\frac{d}{ds}\tilde{b}_A(s)\big|_{s=0} = \int_0^\infty xA(x)dB(x). \tag{2}$$

- The probabilities $\mathbb{P}[B \leq A]$ and $\mathbb{P}[B \geq A]$ are associated with transformations through the following ratios:

$$\tilde{a}_B(0) = \int_0^\infty B(x)dA(x) \quad = \quad \mathbb{P}[B \leq A] \equiv p,$$

$$\tilde{b}_A(0) = \int_0^\infty A(x)dB(x) \quad = \quad \mathbb{P}[B > A] \equiv q = 1 - p.$$

- Note the property of transformations (1)

$$\tilde{a}_{1-B}(s) = \tilde{a}(s) - \tilde{a}_B(s), \quad \tilde{b}_{1-A}(s) = \tilde{b}(s) - \tilde{b}_A(s). \tag{3}$$

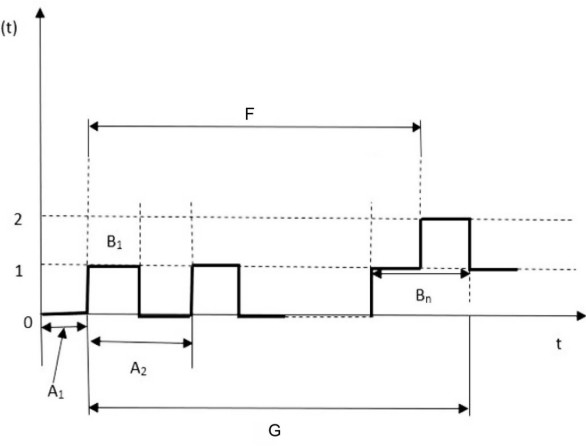

**Figure 2.** Trajectory of the process $J$.

## 3. Reliability Function

Process $J$ is a regenerative one. A trajectory of this process is illustrated in Figure 2.

Here, $F$ stands for the time between system failures, i.e., the period of time the system begins to operate again (state 1) up to the next system failure (state 2). The variable $G$ specifies the length of a regenerative cycle according to a defined renovation policy. The following lemma holds for the LSTs of the time $F$ between failures and the time to the first failure $F_1$.

**Lemma 1.** *The LST* $\tilde{f}(s) = \mathbb{E}\left[e^{-sF}\right]$ *of the time F between failures and the LST* $\tilde{f}_1(s) = \mathbb{E}\left[e^{-sF_1}\right]$ *of the time to the first failure* $F_1$ *are of the form:*

$$\tilde{f}(s) = \frac{\tilde{a}(s) - \tilde{a}_B(s)}{1 - \tilde{a}_B(s)}, \qquad \tilde{f}_1(s) = \tilde{a}(s)\frac{\tilde{a}(s) - \tilde{a}_B(s)}{1 - \tilde{a}_B(s)}. \tag{4}$$

**Proof.** The life time of the system $F$ is a time between two successive failures. After a failure in state 0, the system goes to state 1 where there can be two events: either the recovery of a unit in state 1 in time $B$ and subsequent transition again to state 0 or a failure of a second unit in time $A$ will happen. After a failure in state 1, the system comes back to state 1 in time $B$. Hence, as it can be seen from Figure 2, the system lifetime $F$ satisfies the following stochastic equation:

$$F = \begin{cases} A + F & \text{if } B < A, \\ A & \text{if } B > A. \end{cases} \tag{5}$$

The time to the first system failure $F_1$ satisfies

$$F_1 = A + F. \tag{6}$$

Applying LT to the Equation (5) and taking into account notations (1), one can obtain

$$\tilde{f}(s) = \mathbb{E}\left[e^{-sF}\right] = \int_0^\infty e^{-st}dF(t) =$$

$$= \int_0^\infty e^{-sx}\left[B(x)\tilde{f}(s) + (1 - B(x))\right]dA(x) =$$

$$= \tilde{f}(s)\tilde{a}_B(s) + \tilde{a}_{1-B}(s). \tag{7}$$

Due to the last equation, one can find that the LST of the system lifetime $\tilde{f}(s)$ is given by the first relation in of (4). The second equation of lemma directly follows from stochastic Equation (6). □

**Remark 1.** *Let* $\tilde{\varphi}_i(s), i = 0, 1, 2,$ *be the LST of the distribution of time to the first failure of the system, starting from state i at* $t = 0$. *In previous statement* $\tilde{f}_1(s) = \tilde{\varphi}_0(s)$. *The following equations hold:*

$$\tilde{\varphi}_0(s) = \tilde{q}_{0,1}(s)\tilde{\varphi}_1(s),$$
$$\tilde{\varphi}_1(s) = \tilde{q}_{1,1}(s)\tilde{\varphi}_1(s) + \tilde{q}_{1,2}(s).$$

*Here,* $\tilde{q}_{i,j}(s)$ *stands for the LST of the transition time distribution* $q_{i,j}(t)$ *from state i to state j. Obviously, we have* $\tilde{q}_{0,1}(s) = \tilde{a}(s), \tilde{q}_{1,1}(s) = \tilde{a}_B(s)$ *and* $\tilde{q}_{1,2}(s) = \tilde{a}_{1-B}(s)$. *Expressing* $\tilde{\varphi}_0(s)$, *we get the relation*

$$\tilde{\varphi}_0(s) = \frac{\tilde{q}_{0,1}(s)\tilde{q}_{1,2}(s)}{1 - \tilde{q}_{1,1}(s)},$$

*which coincide with the relation (4) for the first time to failure.*

The main result of this section is the following theorem:

**Theorem 1.** *The LT $\tilde{R}_1(s)$ of the system reliability function $R_1(t) = 1 - F_1(t)$ is*

$$\tilde{R}_1(s) = \frac{(1 - \tilde{a}(s))(1 + \tilde{a}(s) - \tilde{a}_B(s))}{s(1 - \tilde{a}_B(s))}. \tag{8}$$

**Proof.** Taking into account that the LT of any CDF connected with its LST through a relation

$$\tilde{F}_1(s) = \int_0^\infty e^{-st} F_1(t) dt = \frac{1}{s} \int_0^\infty e^{-st} dF_1(t) = \frac{1}{s} \tilde{f}_1(s),$$

the LT $\tilde{R}_1(s)$ of the reliability function $R_1(t) = 1 - F_1(t)$ can be derived using (4) in form

$$\tilde{R}_1(s) = \frac{1}{s} - \tilde{F}_1(s) = \frac{1}{s}(1 - \tilde{f}_1(s)) = \frac{1 - \tilde{a}_B(s) - \tilde{a}^2(s) + \tilde{a}(s)\tilde{a}_B(s)}{s(1 - \tilde{a}_B(s))}$$

$$= \frac{(1 - \tilde{a}(s))(1 + \tilde{a}(s) - \tilde{a}_B(s))}{s(1 - \tilde{a}_B(s))},$$

which ends the proof. $\square$

**Corollary 1.** *The LT $\tilde{R}(s)$ of the survival probability $R(t) = 1 - F(t)$ is of the form*

$$\tilde{R}(s) = \frac{\tilde{R}(s)}{1 + \tilde{a}(s) - \tilde{a}_B(s)} = \frac{1 - \tilde{a}(s)}{s(1 - \tilde{a}_B(s))}, \tag{9}$$

*which follows directly from (4) and (8).*

**Corollary 2.** *The mean system life time between failures $\mathbb{E}[F]$ and mean time to the first failure $\mathbb{E}[F_1]$ are of the form:*

$$\mathbb{E}[F] = \frac{a}{q}, \quad \mathbb{E}[F_1] = a + \frac{a}{q}. \tag{10}$$

**Proof.** The first relation follows from the equality

$$\mathbb{E}[F] = \tilde{R}(0) = \lim_{s \to 0} \frac{1 - \tilde{a}(s)}{s(1 - \tilde{a}_B(s))} = \frac{a}{1 - \tilde{a}_B(0)} = \frac{a}{q},$$

and the second one follows directly from the relation (6). $\square$

## 4. Time Dependent System State Probabilities

### 4.1. Definition

For calculation of the time dependent system state probabilities (TDSSP), we use a renewal theory. In our case, the process $J$ is a regenerative process with a delay (see Figure 2) and its regeneration times are

$$S_0 = 0, \; S_1 = A_1, \; S_2 = S_1 + G_1, \; \ldots, \; S_{k+1} = S_k + G_k, \; \ldots.$$

Here, regeneration cycles $G_i$ $(i = 1, 2, \ldots)$ are the time intervals between two successive returns of the process $J$ into state 1 after a system failure, when one of the system units begins to operate and another one is being repaired. Thus, if the system functioning begins when both units are in operational state, the first regeneration period differs from the others and has the duration $A_1$. The state probabilities of the process $J$ in any time $t$

$$\pi_j(t) = \mathbb{P}\{J(t) = j\}, \quad j = 0, 1, 2$$

can be represented in terms of its distribution over separate regeneration period

$$\pi_j^{(1)}(t) = \mathbb{P}[J(t) = j, \ t < G], \quad j = 0, 1, 2, \tag{11}$$

and the renewal function $H(t)$ can be defined as follows:

$$\pi_i(t) = \pi_i^{(1,1)}(t) + \int_0^t dH(u)\pi_j^{(1)}(t - u). \tag{12}$$

The distribution of the process $J$ on this period is of the form

$$\pi_j^{(1,1)}(t) = \mathbb{P}[J(t) = j, \ t < A_1] = \delta_{j0}(1 - A(t)). \tag{13}$$

In terms of the CDF $G(t) = \mathbb{P}[G_i \le t]$ of RVs $G_i$, a renewal function $H(t)$ is determined as follows:

$$H(t) = \sum_{k \ge 1} \mathbb{P}\left[\left(\sum_{1 \le i \le k} G_i\right) \le t\right] = \sum_{k \ge 1} G^{*k}(t). \tag{14}$$

On the basis of the foregoing, the next lemma can be proved. Consider, first of all, the regeneration cycle distribution.

**Lemma 2.** *The LST of the regeneration cycle is of the form:*

$$\tilde{g}(s) = \mathbb{E}[e^{-sG}] = \frac{\tilde{b}_A(s)}{1 - \tilde{a}_B(s)}. \tag{15}$$

**Proof.** The regeneration cycle is a time between two successive visits of state 1 where two events can occur: either the recovery of a unit in state 1 in time $B$ and subsequent transition to state 0 or a failure of a second unit in time $A$ happens. Figure 2 shows that RV $G$ satisfies the following stochastic equation due to independence of RVs $A$ and $B$:

$$G = \begin{cases} A + G & \text{if } A > B, \\ B & \text{if } A \le B. \end{cases} \tag{16}$$

Applying LST to this stochastic equation, we obtain

$$\begin{aligned} \tilde{g}(s) &= \mathbb{E}\left[e^{-sG}\right] = \int_0^\infty e^{-st} dG(t) \\ &= \int_0^\infty dA(x) \left[B(x)e^{-sx}\tilde{g}(s) + \int_{y>x} e^{-sy} dB(y)\right] = \\ &= \tilde{g}(s) \int_0^\infty e^{-sx} B(x) dA(x) + \int_0^\infty e^{-sy} A(y) dB(y) = \\ &= \tilde{g}(s)\tilde{a}_B(s) + \tilde{b}_A(s), \end{aligned} \tag{17}$$

which implies the expression (15) for the LST of the regeneration period. □

As a corollary, we obtain the mean length of the regeneration period.

**Corollary 3.** *The mean length of the regeneration period can be defined as:*

$$\mathbb{E}[G] = \frac{a_B + b_A}{q}. \tag{18}$$

**Proof.** The proof is obtained by differentiation

$$\mathbb{E}[G] = -\frac{d}{ds}\tilde{g}(s)\big|_{s=0} = -\frac{\tilde{b}'_A(0)(1 - \tilde{a}_B(0)) + \tilde{a}'_B\tilde{b}_A(0)}{(1 - \tilde{a}_B(0))^2}.$$

This expression leads to expression (18) due to properties of $\tilde{a}_B(0)$ and $\tilde{b}_A(0)$.  □

**Lemma 3.** *The LST of the system renewal function is given by:*

$$\tilde{h}(s) = \frac{\tilde{b}_A(s)}{1 - (\tilde{a}_B(s) + \tilde{b}_A(s))}. \tag{19}$$

**Proof.** From the renewal theory, it is well known (and follows from (14)) that the LST of the renewal function $H(t)$ is defined as $\tilde{h}(s) = \tilde{g}(s)(1 - \tilde{g}(s))^{-1}$. Thus, substitution to this expression of (15) for $\tilde{g}(s)$ leads to (19).  □

**Theorem 2.** *The LTs $\tilde{\pi}_j(s)$ of the TDSSPs $\pi_j(t)$ $(j = 0, 1, 2)$ are of the form:*

$$\tilde{\pi}_j(s) = \delta_{j0}\frac{1 - \tilde{a}(s)}{s} + \frac{\tilde{b}_a(s)}{1 - (\tilde{a}_B(s) + \tilde{b}_A(s))}\tilde{\pi}_j^{(1)}(s), \ j = 0, 1, 2, \tag{20}$$

*where $\tilde{\pi}_j^{(1)}(s)$ $(j = 0, 1, 2)$ are the LTs of the TDSSPs $\pi_j^{(1)}(t)$ $(j = 0, 1, 2)$ in a separate regeneration period. These probabilities will be calculated in the next subsection.*

**Proof.** The form of Equations (12) and (14) prompts to calculate the process state probabilities in terms of their LTs and the renewal function LST

$$\tilde{\pi}_j(s) = \int_0^\infty e^{-st}\pi_j(t)dt, \quad \tilde{h}(s) = \int_0^\infty e^{-st}dH(t).$$

Applying LT to Equation (12) and taking into account Equation (13), one can obtain

$$\tilde{\pi}_j(s) = \delta_{j0}\frac{1 - \tilde{a}(s)}{s} + \tilde{h}(s)\tilde{\pi}_j^{(1)}(s). \tag{21}$$

Substitution into this equality of the expression for $\tilde{h}(s)$ (19) leads to (20) and completes the proof.  □

*4.2. The State Probabilities in a Separate Regeneration Period*

Now, we calculate the TDSSPs in a separate regeneration period. The probability $\pi_2^{(1)}(t)$ can be calculated very easily in the main level regeneration period.

**Lemma 4.** *The LT of the second TDSSP $\pi_2^{(1)}(t)$ in the main regeneration period is given by*

$$\tilde{\pi}_2^{(1)}(s) = \frac{\tilde{a}(s) - (\tilde{a}_B(s) + \tilde{b}_A(s))}{s(1 - \tilde{a}_B(s))}. \tag{22}$$

**Proof.** Due to representation of the regeneration period by Formula (16), the event $\{J(t) = 2, \ t < G\}$ occurs if and only if either the event $\{A_1 \le t \le B_1\}$ occurs or the events $\{t > u = A_1 > B_1\}$ and

$\{J(t-u)=2,\ t-u<G\}$ occur. Thus, as it is shown in Figure 2, for the probability $\pi_2^{(1)}(t)$, it holds that

$$\pi_2^{(1)}(t) = \mathbb{P}[J(t)=2,\ t<G] = \mathbb{P}[A \le t < B] + \int_0^t dA(u)B(u)\pi_2^{(1)}(t-u).$$

From this equation for the LST, it follows

$$\tilde{\pi}_2^{(1)}(s) = \int_0^\infty e^{-st}A(t)(1-B(t))dt + \tilde{a}_B(s)\pi_2^{(1)}(s)$$

and

$$\tilde{\pi}_2^{(1)}(s) = \frac{1}{1-\tilde{a}_B(s)}\int_0^\infty e^{-st}A(t)(1-B(t))dt.$$

Calculating LTs of the integral in these expressions using partial integration, we obtain

$$
\begin{aligned}
\tilde{\pi}_2^{(1)}(s) \;=\;& \frac{1}{1-\tilde{a}_B(s)}\int_0^\infty e^{-st}A(t)(1-B(t))dt = \\[2mm]
=\;& -\frac{1}{s(1-\tilde{a}_B(s))}e^{-st}A(t)(1-B(t))dt\Big|_0^\infty + \\[2mm]
+\;& \frac{1}{s(1-\tilde{a}_B(s))}\int_0^\infty e^{-st}[dA(t)(1-B(t))-A(t)dB(t)] = \\[2mm]
=\;& \frac{\tilde{a}(s)-(\tilde{a}_B(s)+\tilde{b}_A(s))}{s(1-\tilde{a}_B(s))}
\end{aligned}
$$

that completes the proof.　□

Since the calculation of the probabilities $\pi_j^{(1)}$ $(j=0,1)$ is not a trivial task, we intend to apply the theory of DSRP [3,8,12]. According to this theory, the further analysis can be described as follows. As it was shown in the Equation (12), the process distribution can be represented in terms of its distribution in embedded regeneration cycles and embedded renewal function. This approach is useful if the behavior of the process is too complex in order to derive a state distribution directly, but the embedded regeneration epochs are available. At known regeneration epochs $S_k^{(1)}$, the distribution in the basic regeneration period (regeneration period of first level) $\pi_i^{(1)}(t)$ similar to the Equation (12) can be represented in terms of distributions in embedded regeneration cycles (regeneration cycles of the second level) $\pi_j^{(2)}(t)$ and embedded renewal function $H^{(1)}(t)$ in the following way:

$$\pi_j^{(1)}(t) = \pi_j^{(2)}(t) + \int_0^\infty dH^{(1)}(u)\pi_j^{(2)}(t-u),\quad j=0,1, \tag{23}$$

where

$$\pi_j^{(2)}(t) = \mathbb{P}\{J(t)=j,\ t<G^{(2)}\},\quad j=0,1$$

is the process state distribution in a separate regeneration period of the second level. In this case as embedded regeneration epochs, we choose the random number $\nu=\min\{n:\ A_n<B_n\}$ of time epochs

$$S_1^{(1)} = A_1 1_{\{A_1>B_1\}},\quad S_2^{(1)} = S_1^{(1)} + A_2 1_{\{A_1>B_1,\ A_2>B_2\}},\dots$$

up to time, when the event $\{A_n \le B_n\}$ occurs for the first time. It means that the time interval $G^{(1)}$ between embedded regeneration epochs has a distribution $G^{(1)}(t)=A(t)$ and these epochs in the time

interval $F$ are determined by Equation (5). According to the theory of DSRP, the embedded renewal function $H^{(1)}(t)$ satisfies the equation

$$H^{(1)}(t) = A(t) + \int_0^t dH^{(1)}(u)A(t-u) - F(t),\tag{24}$$

where $F(t)$ is a CDF of the time between system failures determined by its LST (4).

Similar to the basic case, the solution of Equations (23) and (24) can be represented in terms of their LTs and LSTs:

$$\tilde{\pi}_j^{(2)}(s) = \int_0^\infty e^{-st}\pi_j^{(2)}(t)dt, \quad \tilde{h}^{(1)}(s) = \int_0^\infty e^{-st}dH^{(1)}(t).$$

The next lemma specifies connections between process distributions in the first and in the second level regeneration cycles in terms of their LTs.

**Lemma 5.** *The TDSSPs in the first and in the second level regeneration cycles in terms of the LTs satisfy the relation*

$$\tilde{\pi}_j^{(1)}(s) = \tilde{\pi}_j^{(2)}(s)\frac{\tilde{a}_B(s)}{1 - \tilde{a}_B(s)} \quad j = 0, 1.\tag{25}$$

**Proof.** Applying LT to Equation (23), we get

$$\tilde{\pi}_j^{(1)}(s) = (1 + \tilde{h}^{(1)}(s))\tilde{\pi}_j^{(2)}(s),\tag{26}$$

where, due to (24), the LST $\tilde{h}^{(1)}(s)$ of the embedded renewal function $H^{(1)}(t)$ is of the form

$$\tilde{h}^{(1)}(s) = \tilde{a}(s) + \tilde{h}^{(1)}(s)\tilde{a}(s) - \tilde{f}(s),$$

which leads in turn to

$$\tilde{h}^{(1)}(s) = \frac{\tilde{a}(s) - \tilde{f}(s)}{1 - \tilde{a}(s)}.$$

Substitution of this relation to the Equation (26) and taking into account expressions for $\tilde{f}(s)$ from (4) implies the result (22) that completes the proof. □

We have to calculate now only $\tilde{\pi}_j^{(2)}(s) \ (j = 0, 1)$.

**Lemma 6.** *In notations (4), the LTs of the second level system state probabilities are:*

$$\begin{aligned}
\tilde{\pi}_0^{(2)}(s) &= \frac{1}{s}(\tilde{a}(s) - (\tilde{a}_B(s) + \tilde{b}_A(s))); \\
\tilde{\pi}_1^{(2)}(s) &= \frac{1}{s}[1 - (\tilde{a}(s) + \tilde{b}(s)) + (\tilde{a}_B(s) + \tilde{b}_A(s))].
\end{aligned}\tag{27}$$

**Proof.** First, consider appropriate expressions for $\tilde{\pi}_j^{(2)}(t) \ (j = 0, 1)$. From Figure 2, it follows that

- The event $\{J(t) = 0, \ t < G^{(1)}\}$ occurs if and only if $\{B \le t < A\}$;
- The event $\{J(t) = 1, \ t < G^{(1)}\}$ occurs if and only if $\{t < B \le A\}$ or $\{t < A \le B\}$;

Hence, the appropriate probabilities are

$$
\begin{aligned}
\pi_0^{(2)}(t) &= \mathbb{P}[B < t < A] = B(t)(1 - A(t)), \\
\pi_1^{(2)}(t) &= \mathbb{P}[t < B < A] + \mathbb{P}[t < A < B] = \\
&= \int_t^\infty (1 - A(u))dB(u) + \int_t^\infty (1 - B(u))dA(u).
\end{aligned}
\tag{28}
$$

Calculating LTs of these expressions by partial integration in terms of notations (1) gives

$$
\begin{aligned}
\tilde{\pi}_0^{(2)}(s) &= \int_0^\infty e^{-st} B(t)(1 - A(t))dt = \\
&= \frac{1}{s} \int_0^\infty e^{-st}[dB(t)(1 - A(t)) - B(t)dA(t)] = \\
&= \frac{1}{s}(\tilde{b}(s) - (\tilde{a}_B(s) + \tilde{b}_A(s))); \\
\tilde{\pi}_1^{(2)}(s) &= \int_0^\infty e^{-st} \int_t^\infty [(1 - A(u))dB(u) + (1 - B(u))dA(u)] = \\
&= \int_0^\infty \int_0^u e^{-st} dt\, [(1 - A(u))dB(u) + (1 - B(u))dA(u)] = \\
&= \frac{1}{s} \int_0^\infty (1 - e^{-su}) [(1 - A(u))dB(u) + (1 - B(u))dA(u)] = \\
&= \frac{1}{s}[1 - (\tilde{a}(s) + \tilde{b}(s)) + (\tilde{a}_B(s) + \tilde{b}_A(s))]
\end{aligned}
$$

that ends the proof.  □

By combining all obtained results for LTs of the TDSSPs, one can get the following statement.

**Theorem 3.** *The LTs of the TDSSPs are of the form:*

$$
\begin{aligned}
\tilde{\pi}_0(s) &= \frac{1 - \tilde{a}(s)}{s} + \frac{\tilde{b}_A(s)}{s(1 - \tilde{a}_B(s))} \times \frac{\tilde{b}(s) - (\tilde{a}_B(s) + \tilde{b}_A(s))}{1 - (\tilde{a}_B(s) + \tilde{b}_A(s))}; \\
\tilde{\pi}_1(s) &= \frac{\tilde{b}_A(s)}{s(1 - \tilde{a}_B(s))} \times \frac{1 - (\tilde{a}(s) + \tilde{b}(s)) + (\tilde{a}_B(s) + \tilde{b}_A(s))}{1 - (\tilde{a}_B(s) + \tilde{b}_A(s))}; \\
\pi_2(s) &= \frac{\tilde{b}_A(s)}{s(1 - \tilde{a}_B(s))} \times \frac{\tilde{a}(s) - (\tilde{a}_B(s) + \tilde{b}_A(s))}{1 - (\tilde{a}_B(s) + \tilde{b}_A(s))}.
\end{aligned}
\tag{29}
$$

**Proof.** For $j = 0, 1$, the proof is presented as a series of equalities:

$$
\begin{aligned}
\tilde{\pi}_j(s) &= \delta_{j0} \frac{1 - \tilde{a}(s)}{s} + \tilde{h}(s)\pi_j^{(1)}(s) = \\
&= \delta_{j0} \frac{1 - \tilde{a}(s)}{s} + \frac{\tilde{g}(s)}{1 - \tilde{g}(s)}(1 + \tilde{h}_j^{(1)}(s))\pi_j^{(2)}(s) = \\
&= \delta_{j0} \frac{1 - \tilde{a}(s)}{s} + \frac{\tilde{b}_A(s)}{1 - (\tilde{a}_B(s) + \tilde{b}_A(s))} \frac{\tilde{a}(s) - \tilde{f}(s)}{1 - \tilde{a}(s)}\pi_j^{(2)}(s) = \\
&= \delta_{j0} \frac{1 - \tilde{a}(s)}{s} + \frac{\tilde{b}_A(s)}{1 - (\tilde{a}_B(s) + \tilde{b}_A(s))} \frac{1}{1 - \tilde{a}_B(s)}\pi_j^{(2)}(s).
\end{aligned}
$$

Substitution of expressions (27) for $\tilde{\pi}_j^{(2)}(s)$ leads to the first two formulas from (29). For the last one, it holds that

$$\tilde{\pi}_2(s) = \tilde{h}(s)\tilde{\pi}_2^{(1)}(s) = \frac{\tilde{g}(s)}{1-\tilde{g}(s)}\tilde{\pi}_2^{(1)}(s) = \frac{\tilde{b}_A(s)}{1-(\tilde{a}_B(s)+\tilde{b}_A(s))}\tilde{\pi}_2^{(1)}(s).$$

Substitution into this formula of expression (22) for $\tilde{\pi}_2^{(1)}(s)$ leads to the last formula from (29).    □

## 5. Stationary State Probabilities

To calculate the stationary state probabilities (SSSPs), we use a Tauber theorem. Due to (29), we get

$$\pi_j = \lim_{t\to\infty}\pi_j(t) = \lim_{s\to 0} s\tilde{\pi}_j(s). \tag{30}$$

By means of this formula, we can prove the next statement.

**Theorem 4.** *The system state stationary probabilities are:*

$$\begin{aligned}
\pi_0 &= 1 - \frac{b}{a_B+b_A}, \\
\pi_1 &= \frac{a+b}{a_B+b_A} - 1, \\
\pi_2 &= 1 - \frac{a}{a_B+b_A}.
\end{aligned} \tag{31}$$

**Proof.** According to the limit relation (30), we obtain

$$\begin{aligned}
\pi_0 &= \lim_{s\to 0}\left[1-\tilde{a}(s)+\frac{\tilde{b}_A(s)}{1-\tilde{a}_B(s)}\times\frac{\tilde{b}(s)-(\tilde{a}_B(s)+\tilde{b}_A(s))}{1-(\tilde{a}_B(s)+\tilde{b}_A(s))}\right] = \\
&= \lim_{s\to 0}\frac{\tilde{b}(s)-(\tilde{a}_B(s)+\tilde{b}_A(s))}{1-(\tilde{a}_B(s)+\tilde{b}_A(s))}; \\
\pi_1 &= \lim_{s\to 0}\left[\frac{\tilde{b}_A(s)}{1-\tilde{a}_B(s)}\times\frac{1-(\tilde{a}(s)+\tilde{b}(s))+(\tilde{a}_B(s)+\tilde{b}_A(s))}{1-(\tilde{a}_B(s)+\tilde{b}_A(s))}\right]; \\
&= \lim_{s\to 0}\frac{1-(\tilde{a}(s)+\tilde{b}(s))+(\tilde{a}_B(s)+\tilde{b}_A(s))}{1-(\tilde{a}_B(s)+\tilde{b}_A(s))}; \\
\pi_2 &= \lim_{s\to 0}\frac{\tilde{b}_A(s)}{1-\tilde{a}_B(s)}\times\frac{\tilde{a}(s)-(\tilde{a}_B(s)+\tilde{b}_A(s))}{1-(\tilde{a}_B(s)+\tilde{b}_A(s))} = \\
&= \lim_{s\to 0}\frac{\tilde{a}(s)-(\tilde{a}_B(s)+\tilde{b}_A(s))}{1-(\tilde{a}_B(s)+\tilde{b}_A(s))}.
\end{aligned}$$

Using L'Hopital's rule, we get the next

$$\begin{aligned}
\pi_0 &= \frac{\tilde{b}'(0)-(\tilde{a}_B'(0)+\tilde{b}_A'(0))}{-(\tilde{a}_B'(0)+\tilde{b}_A'(0))} = 1-\frac{b}{a_B+b_A}; \\
\pi_1 &= \frac{-(\tilde{a}(0)+\tilde{b}'(0))+(\tilde{a}_B'(0)+\tilde{b}_A'(0))}{-(\tilde{a}_B'(0)+\tilde{b}_A'(0))} = \frac{a+b}{a_B+b_A}-1; \\
\pi_2 &= \frac{\tilde{a}'(0)-(\tilde{a}_B'(0)+\tilde{b}_A'(0))}{-(\tilde{a}_B'(0)+\tilde{b}_A'(0))} = 1-\frac{a}{a_B+b_A}
\end{aligned}$$

that completes the proof.    □

**Example 1.** *As an example, consider a Markov model* $< M_2|M|1 >$, *when*

$$A(t) = 1 - e^{-\alpha t}, \quad B(t) = 1 - e^{-\beta t}.$$

*For the SSSPs, we get in terms of parameter* $\rho = \frac{\alpha}{\beta}$:

$$a_B + b_A = \left[\frac{1}{\alpha} + \frac{1}{\beta} - \frac{1}{\alpha + \beta}\right] = \frac{\alpha^2 + \alpha\beta + \beta^2}{\alpha\beta(\alpha + \beta)}$$

*Now, for* $\pi_j$ $(j = 0, 1, 2)$, *it holds that*

$$
\begin{aligned}
\pi_0 &= 1 - \frac{b}{a_b + b_A} = 1 - \frac{\alpha(\alpha + \beta)}{\alpha^2 + \alpha\beta + \beta^2} = \frac{1}{1 + \rho + \rho^2}; \\
\pi_1 &= \frac{a + b}{a_B + b_A} - 1 = \frac{\alpha\beta}{\alpha^2 + \alpha\beta + \beta^2} = \frac{\rho}{1 + \rho + \rho^2}; \\
\pi_2 &= 1 - \frac{a}{a_B + b_A} = \frac{\alpha^2}{\alpha^2 + \alpha\beta + \beta^2} = \frac{\rho^2}{1 + \rho + \rho^2}.
\end{aligned}
$$

*This result coincides with those calculated by direct approach using Birth and Death process for the Markov case.*

## 6. Numerical Analysis

In this section, we study the sensitivity of reliability function, TDSSPs and SSSPs to types of distribution functions and their parameters. The life- and repair-times will follow exponential, PH-type, Pareto and log-normal distributions. For valid comparison of the results, we choose the parameters of these distributions in such a way that the mean and the variance of the life and repair times always coincide with an exponential version, i.e., $\mathbb{E}[A] = a = \frac{1}{\alpha}$ and $\mathbb{E}[B] = b = \frac{1}{\beta}$. Now, we give a short description of the considered life-time distributions $A(t)$ with appropriately selected parameters. The repair-time distributions $B(t)$ are defined by analogy.

The PH-type distribution $\mathcal{PH}$ with representation $(\boldsymbol{\eta}, M)$,

$$A(t) = 1 - \boldsymbol{\eta}e^{Mt}\mathbf{1}, \; \mathbb{E}[A] = a = -\boldsymbol{\eta}M^{-1}\mathbf{1} = \frac{1}{\alpha}, \; \mathbb{V}[A] = 2\boldsymbol{\eta}M^{-2}\mathbf{1} - a^2 = \frac{1}{\alpha^2},$$

where $\boldsymbol{\eta} = (0.7, 0.1, 0.2)$ and

$$
M = \begin{pmatrix} -0.80 & 0.19 & 0.50 \\ 3.35 & -3.50 & 0.10 \\ 0.80 & 0.60 & -1.50 \end{pmatrix} \text{ for } a = 10, \quad
M = \begin{pmatrix} -2.80 & 1.14 & 1.00 \\ 13.39 & -15.00 & 1.50 \\ 2.80 & 1.60 & -4.50 \end{pmatrix} \text{ for } a = 2,
$$

$$
M = \begin{pmatrix} -8.00 & 1.94 & 5.00 \\ 33.48 & -35.00 & 1.00 \\ 8.00 & 6.00 & -15.00 \end{pmatrix} \text{ for } a = 1, \quad
M = \begin{pmatrix} 16.00 & 1.03 & 10.00 \\ 25.43 & -35.00 & 2.00 \\ 16.00 & 10.00 & -30.00 \end{pmatrix} \text{ for } a = 0.2,
$$

$$
M = \begin{pmatrix} -35.00 & 18.81 & 5.00 \\ 94.70 & -110.00 & 10.00 \\ 10.00 & 60.00 & -80.00 \end{pmatrix} \text{ for } b = 0.1.
$$

The Pareto distribution $\mathcal{PAR}$ with parameters $(k, t_{\min})$,

$$A(t) = 1 - \left(\frac{t_{\min}}{t}\right)^k, \; t \geq t_{\min}, \; k = 1 + \sqrt{2}, \; t_{\min} = \frac{\sqrt{2}}{\alpha(1 + \sqrt{2})}.$$

The log-normal distribution $\mathcal{LN}$ with parameters $(\mu, \sigma)$,

$$A(t) = \Phi\left(\frac{\ln(x) - \mu}{\sigma}\right), \mu = \ln\frac{1}{\alpha} - \frac{\ln(2)}{2}, \sigma = \sqrt{\ln(2)}.$$

Now, we fix the parameters of the repair-time distributions to guarantee that $\mathbb{E}[B] = b = 0.1$ and $\mathbb{V}[B] = 0.01$. In Figure 3, we demonstrate the reliability $R_1(t)$ and survival $R(t)$ functions obtained by numerical inversion with Euler algorithm of the corresponding LTs (8) and (9) in case $A \sim \mathcal{PH}(\eta_1, M_1)$ and $B \sim \mathcal{PH}(\eta_2, M_2)$. Obviously, for $a \geq 1$, we observe that the functions exhibit a heavier tail. Note that these graphics are indistinguishable from those obtained for exponential distributions.

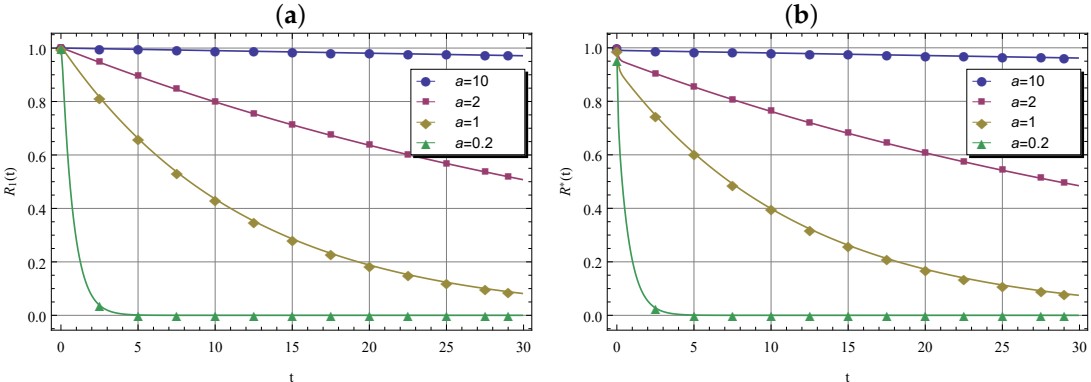

**Figure 3.** Functions $R_1(t)$ (**a**) and $R(t)$ (**b**) for $A \sim \mathcal{PH}(\eta_1, M_1)$ and $B \sim \mathcal{PH}(\eta_1, M_1)$ versus $a$.

In case it is not feasible to evaluate Laplace–Stieltjes transforms in explicit form (4), different approaches can be used instead to recover the distribution function via moments. We have tried the maximum entropy solution and simple fitting by a mixture of exponential distributions. Both of these methods work and give reasonable approximations. For details about maximum entropy solution, the reader is referred to Falin et al. [22]. To illustrate this methodology, we limit ourselves to analyzing the length of the system life time between failures $F$, when the first, e.g., four moments $\mathbb{E}[F^k] = (-1)^k \frac{d^k}{ds^k} \tilde{f}(s)\big|_{s=0}, 1 \leq k \leq m = 4$, are available,

$$\mathbb{E}[F] = \frac{a}{q},$$

$$\mathbb{E}[F^2] = \frac{2a \cdot a_B + qa^{(2)}}{q^2},$$

$$\mathbb{E}[F^3] = \frac{3a(qa_B^{(2)} + 2a_B^2) + q(3a^{(2)}a_B + qa^{(3)})}{q^3},$$

$$\mathbb{E}[F^4] = \frac{4a(6a_B^2 + 6qa_Ba_B^{(2)} + q^2a_B^{(3)}) + q(12a^{(2)}a_B^2 + 4qa^{(3)}a_B + q(6a^{(2)}a_B^{(2)} + qa^{(4)}))}{q^4},$$

where $a^{(k)} = \mathbb{E}[A^k], a_B^{(k)} = \int_0^\infty x^k B(x)dA(x)$. From the maximum entropy formalism, it is known that the maximum entropy solution is of the form

$$\hat{f}(t) := \hat{f}(t; m) = e^{-c_0 - \sum_{k=1}^m c_k t^k}, t > 0 \tag{32}$$

with $c_0 = \ln \int_0^1 e^{-\sum_{k=1}^m c_k t^k} dt$. The Lagrangian coefficients $c_k, 1 \leq k \leq m$, can be calculated as a solution of the following minimization problem:

$$g(c_1, \dots, c_m) = \ln \int_0^\infty e^{-\sum_{k=1}^m c_k(t^k - \mathbb{E}[F^k])} dt.$$

To solve this problem and obtain optimal $c_k^*$, we use Nelder and Mead's algorithm. This direct search method doesn't use derivatives and hence can be used also in case of an almost singular Hessian of $g(c_1, \ldots, c_m)$.

In Figure 4, we plot the maximum entropy solution $\hat{R}(t)$ for the function $R(t)$ in the case when RV $A$ und $B$ either simultaneously $A \sim \mathcal{LN}(\mu_1, \sigma_1)$ and $B \sim \mathcal{LN}(\mu_2, \sigma_2)$ or $A \sim \mathcal{PH}(\boldsymbol{\eta}_1, M_1)$ and $B \sim \mathcal{PH}(\boldsymbol{\eta}_2, M_2)$. We see that, in the log-normal case, the functions exhibit heavier tails by increasing of the mean value $a$s. Figures 3a and 4b can be used to compare the maximum entropy solution with a true function in the case of a PH-type distribution. We may notice no sufficient difference in graphics of functions.

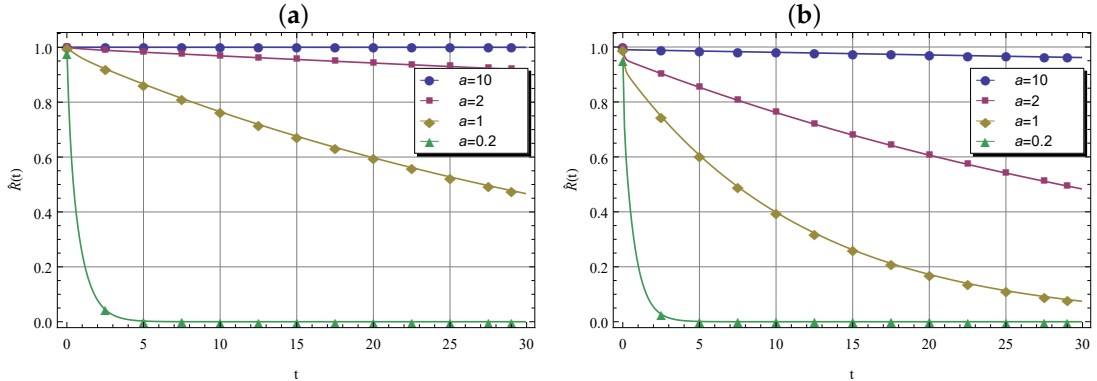

**Figure 4.** Functions $\hat{R}(t)$ for $\mathcal{LN}$ (**a**) and $\mathcal{PH}$ (**b**) distributions versus $a$.

Figures 5 and 6 provide the TDSSPs for the PH-type distributed life- and repair-times for different values of $a$. It should be noticed that the state probability function $\pi_0(t)$ exhibits a concave structure with a corresponding minimum which is shifted to the left as $a$ decreases. The function $\pi_0(t)$ in Figure 5a is shown only up to its minimum but further it increases as well and converges to a stationary value. In addition, we can see how fast the function $\pi_i(t)$ converges to a stationary value $\pi_i$ as $a$ decreases. We strongly believe that the proposed above methodology of distribution recovery via moments will also be helpful for evaluation of the TDSSPs, when there are no closed forms of LSTs for life- and repair-time distributions.

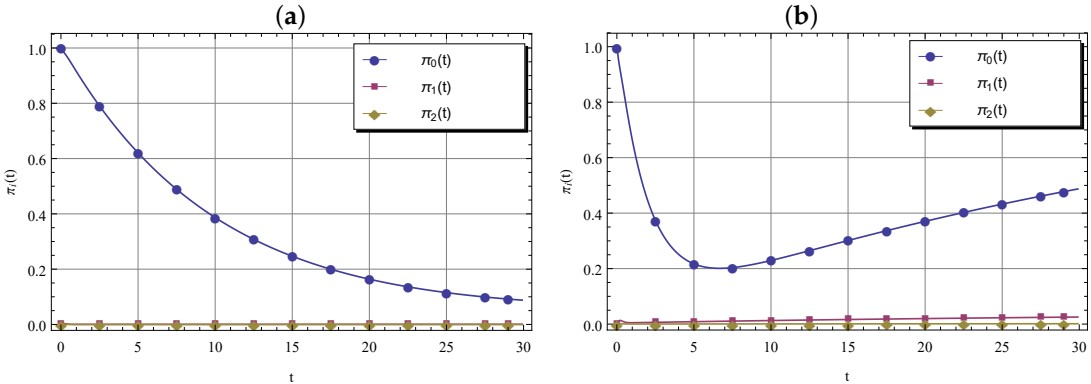

**Figure 5.** TDSSPs $\pi_i(t)$, $i = 0, 1, 2$ for $a = 10$ (**a**) and for $a = 2$ (**b**).

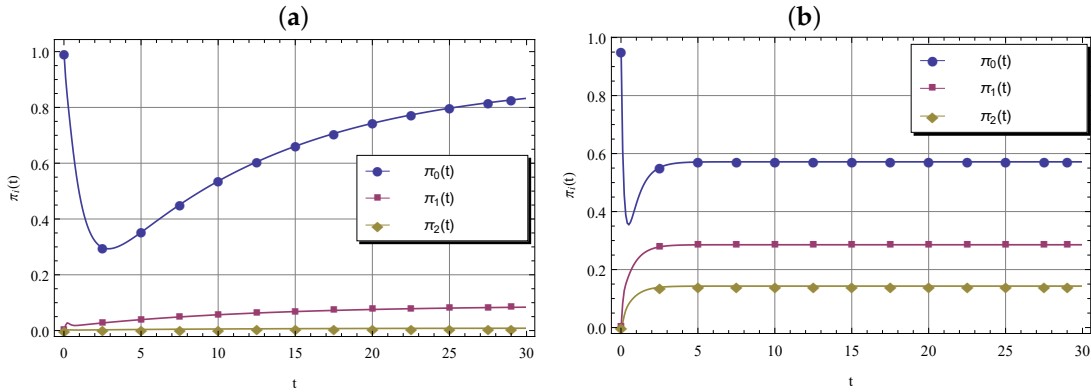

**Figure 6.** TDSSPs $\pi_i(t)$, $i = 0, 1, 2$ for $a = 1$ (**a**) and for $a = 0.2$ (**b**).

The results of SSSP and mean lifetime calculation for different types of distribution pairs $A(t)$ and $B(t)$ are summarized in Tables 1–5. In all cases, the parameters were selected for mean and variance value constraints, i.e., $a = \mathbb{E}[A] = \frac{1}{\alpha}$, $b = \mathbb{E}[B] = \frac{1}{\beta}$, $\mathbb{V}[A] = \frac{1}{\alpha^2}$ and $\mathbb{V}[A] = \frac{1}{\beta^2}$. In experiments demonstrated in Table 1, the life- and repair time distributions have the same distributions but with different parameters fitting to $a = 1$ and $b = 0.1$. High mean values for the time to failure and the time between failures are a consequence of the Pareto distribution property that is defined for $t \geq t_{\min}$.

**Table 1.** $\pi_i$, $i = 0, 1, 2$, $\mathbb{E}[F]$ and $\mathbb{E}[F_1]$ versus $A(t)$ and $B(t)$.

| $A(t)$, $B(t)$ | $\pi_0$ | $\pi_1$ | $\pi_2$ | $\mathbb{E}[F]$ | $\mathbb{E}[F_1]$ |
|---|---|---|---|---|---|
| $\mathcal{E}$ | 0.9009 | 0.0900 | 0.0090 | 11.0000 | 12.0000 |
| $\mathcal{PH}$ | 0.9009 | 0.0901 | 0.0090 | 11.0051 | 12.0051 |
| $\mathcal{PAR}$ | 0.9001 | 0.0989 | 0.0010 | 519.0910 | 520.0910 |
| $\mathcal{LN}$ | 0.9003 | 0.0966 | 0.0031 | 39.5976 | 40.5976 |

The exponential and PH-type distributions generate almost indistinguishable results. We also notice that the distributions with heavier tails decrease the probability of two operational units.

Tables 2–5 present calculations for the systems with different pairs of life- and repair times distributions. The results for exponential and PH-type life type distributions in Tables 2 and 3 are very similar. In addition, there are no sufficient differences in obtained results between different repair time distributions.

**Table 2.** $\pi_i$, $i = 0, 1, 2$, $\mathbb{E}[F]$ and $\mathbb{E}[F_1]$, $A \sim \mathcal{E}(\alpha)$ versus $B(t)$.

| $B(t)$ | $\pi_0$ | $\pi_1$ | $\pi_2$ | $\mathbb{E}[F]$ | $\mathbb{E}[F_1]$ |
|---|---|---|---|---|---|
| $\mathcal{PH}$ | 0.9009 | 0.0901 | 0.0090 | 11.0000 | 12.0000 |
| $\mathcal{PAR}$ | 0.9007 | 0.0919 | 0.0073 | 10.7972 | 11.7972 |
| $\mathcal{LN}$ | 0.9009 | 0.0903 | 0.0088 | 10.9739 | 11.9739 |

**Table 3.** $\pi_i$, $i = 0, 1, 2$, $\mathbb{E}[F]$ and $\mathbb{E}[F_1]$, $A \sim \mathcal{PH}(\eta_1, M_1)$ versus $B(t)$.

| $B(t)$ | $\pi_0$ | $\pi_1$ | $\pi_2$ | $\mathbb{E}[F]$ | $\mathbb{E}[F_1]$ |
|---|---|---|---|---|---|
| $\mathcal{E}$ | 0.9009 | 0.0901 | 0.0090 | 11.0050 | 12.0050 |
| $\mathcal{PAR}$ | 0.9007 | 0.0919 | 0.0073 | 11.0050 | 12.0050 |
| $\mathcal{LN}$ | 0.9009 | 0.0903 | 0.0088 | 11.0050 | 12.0050 |

As it is shown in Tables 4 and 5, the heavy-tailed Pareto and log-normal life-time distributions are a cause for reduction of the system failure probability and increase the mean time to failure and

the mean time between failures. Hence, we notice that reliability and probabilistic characteristics of the system are more sensitive to changing the form of distribution functions when they exhibit heavier tails.

**Table 4.** $\pi_i$, $i = 0, 1, 2$, $\mathbb{E}[F]$ and $\mathbb{E}[F_1]$, $A \sim \mathcal{PAR}(k, t_{\min})$ versus $B(t)$.

| $B(t)$ | $\pi_0$ | $\pi_1$ | $\pi_2$ | $\mathbb{E}[F]$ | $\mathbb{E}[F_1]$ |
|---|---|---|---|---|---|
| $\mathcal{E}$ | 0.9000 | 0.0993 | 0.0001 | 1297.1700 | 1298.1700 |
| $\mathcal{PH}$ | 0.9000 | 0.0999 | 0.0001 | 1299.5000 | 1300.5000 |
| $\mathcal{LN}$ | 0.9001 | 0.0994 | 0.0005 | 452.3640 | 453.3640 |

**Table 5.** Case 5: $\pi_i$, $i = 0, 1, 2$, $\mathbb{E}[F]$ and $\mathbb{E}[F_1]$, $A \sim \mathcal{LN}(\mu, \sigma)$ versus $B(t)$.

| $B(t)$ | $\pi_0$ | $\pi_1$ | $\pi_2$ | $\mathbb{E}[F]$ | $\mathbb{E}[F_1]$ |
|---|---|---|---|---|---|
| $\mathcal{E}$ | 0.9003 | 0.0969 | 0.0028 | 35.4011 | 36.4011 |
| $\mathcal{PH}$ | 0.9003 | 0.0969 | 0.0028 | 35.3988 | 36.3988 |
| $\mathcal{PAR}$ | 0.9002 | 0.0973 | 0.0024 | 59.4631 | 60.4631 |

## 7. Conclusions

We provide probabilistic and reliability analysis of a cold double redundant renewable system with generally distributed life and repair times. We derive in closed-form the reliability function, the time-dependent as well as stationary system state distributions using LSTs and LTs. Numerical results illustrate sensitivity and effect of distribution functions and their parameters on reliability and probabilistic measures of the system. The presented system can be made preventive maintainable. We believe that the corresponding maintenance policy can be effective and leads to an increase in the system reliability.

**Author Contributions:** Conceptualization, V.R. and D.E.; methodology, V.R., D.E. and J.S.; software, D.E. and N.S.; validation, D.E. and N.S.; formal analysis, V.R., D.E. and N.S.; investigation, V.R. and D.E.; writing—original draft preparation, V.R. and D.E.; writing—review and editing, D.E. and N.S.; supervision, D.E.; project administration, D.E. and N.S. All authors have read and agreed to the published version of the manuscript.

**Funding:** The publication has been prepared with the support of the "RUDN University Program 5-100" (V.R. and D.E.: Mathematical model development, investigation, methodology, validation). The reported study was funded by RFBR, project number 17-01-00633 (V.R.: Conceptualization and formal analysis).

**Acknowledgments:** The authors acknowledge, with gratitude, the constructive comments and suggestions of anonymous referees and the Editor.

**Conflicts of Interest:** The authors declare no conflict of interest.

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
