# Peer review of "On Reliability of a Double Redundant Renewable System with a Generally Distributed Life and Repair Times"

_mathematics, doi:10.3390/math8020278_

Round 1

Reviewer 1 Report

Review to the paper

«On reliability of a double redundant renewable system with generally distributed life and repair times”

by Vladimir Rykov, Dmitry Efrosinin, Natalia Stepanova and Janos Sztrik

          The paper is devoted to investigation of reliability of systems mentioned in its title. Wherein the method of imbedded regenerative processes, proposed and developed before by one of the authors of the paper is used. The method is based on construction of some embedded regeneration points and so called embedded renewal equation solution. In considered case this equation represented by the formula (24). This equation intuitively is clear, but may be needed in more detailed explanation.

Another technical remarks:

In the page 13 in all formulas for A(t) intervals between A(t), and its parameters are needed.

At the same page 13, it seems that instead of R^*(t) the value R_1(t) is needed.

At the same page 13, line 151. It is shown that the distributions $A \in PH(\eta_1, M_1 )$ and $B \in PH(\eta_1, M_1 )$ have the same parameters. It is nonsense.

At the same page 13. Figures 2 (a) and (b) demonstrate that all probabilities converge to the value zero, but it is impossible because their sum equal to 1. May be needed to explain this type of the function behavior of the scale of the figure should be changed in order to demonstrate the function behavior for large values of time t.

At the same page 13. For title of Figure 2 may be better to write … “for different values of parameter “a” instead of “versus a”.

Correct References as follows:

Cinlar, E. On semi-Markov processes on arbitrary space; Proc. Cambridge Philos, Mathematical Proceedings of the Cambridge Philosophical Society, 66 (2), pp. 381-392, 1969. Falin,G. I., Martìn Dìaz, M., Artalejo, J. R. (1994). Information theoretic approximations for the M/ G/I retrial queue. Acta Inf 1994, 31, 559–571. Gaver, D. P. (1963) Time to failure and availability of paralleled redundant systems with repair. IEEE Trans. Reliability, R-12, 30 - 38. Jacod, J. Theoreme de renouvelltment et classification pour les chaines semi-Markoviennes. Ann. Inst. Henri Poincare, sect. B, 7 (2), 1971, 83–129. Klimov, G.P. Probability theory and mathematical statistics.; Moscow State University: Moscow, Russia, 1983 (in Russian). Mazumdar, M. Reliability of two-unit redundant repairable systems when failures are revealed by inspections. SIAM J. Appl. Math. 1970, 19, 637–647. Nummelin, E. Uniform and ratio-limit theorems for Markov-renewal and semi-regenerative processes on a general state space. Ann. Inst. Henri Poincare, sect. B, 1978, 14 (2), 1978, 119–143. Rykov, V., Jolkoff, S. Generalized regenerative processes with embedded regeneration periods and their applications. Mathematische Operationsforschung und Statistik. Ser. Optimization 1981, 12, 575–591. Rykov, V. Decomposable semi-regenerative processes and their applications; LAMPERT Academic Publishing, 2011. Rykov, V. On Reliability of Renewable Systems. In Reliability Engineering. Theory and Applications, Vonta I, Ram M (Eds), 2018, 173–196.

Remove reference 11. Check references 14, 16, 17, 18

Author Response

Dear reviewer, thank you for your comments.

Parameters of the PH-type distribution were given on page 13. Notations for reliability function on page 13 were corrected. Representations for PH-type distributions were corrected. The figure 2(a) was additionally explained. We think that "versus a" is better, since it is compact and reflects what this figure illustrates . References was corrected according to requirements of the journal.

Reviewer 2 Report

I disagree with this statement. Reliability comes from Industrial Engineering, and the associated theory has been developed as a need to deal with RAMS aspects (Reliability, Availability, Maintainability, Safety). "Reliability theory is a branch of Operations research which is a discipline that studies applications of foremost analytical methods with the aim to get solutions of different problems."

From a industrial perspective, it is not alwasy easy to link the described variables to real application fields. Considering the scope of the journal, this is not a fully necessary point, but I do recommned to provide a list of variables and parameters at the beginning of the paper.

Consider giving more descriptive details on the "trajectory of the process" described in Figure 1.

The work appears to be accurate, but I leave the detailed revision on mathematical aspects to other reviewers. What is lacking from my perspective is some type of transposition to operational aspects. For example, how could the proposed approach be integrated with reliability issues? How could the proposed function be integrated with inventory management solutions? I think such aspects should be presented in the conclusion which currently is just another (if one counts the abstarct) summary of the paper. Try exploring methodologies well-establshed in the maintenance field, which seems comparable to what has been proposed here, (e.g.) the Multi-Echelon Technique for Recoverable Item Control, and its recent theories also related to cold standby relaibility management.
Besides this point, consider adding a frank account of the limitation of the proposed study, and an overview of potential future research paths.

Author Response

Dear reviewer,

thank you very much for useful comments and remarks.

  1. We have rewritten the first paragraph in the introduction.
  2. The variables and parameters are described as they appear in the text. We could add a description at the beginning of the article, but we think it's redundant. 
  3. We have added a detailed caption for Figure 1.
  4. We agree, it is an interesing task to add some controllable element to the model, e.g. preventive maintenance repair. We have included this remark to the conclusion.
  5. The multi-echelon context is used for optimal inventory modeling and is  Đłuseful for the systems with multiple non-reliable repairable heterogeneous units . It is interesting task and we take this remark into account  by analyzing multiple unit redundant system.

Reviewer 3 Report

The main contribution of the paper consists in the derivation of several probabilistic characteristics for a specific queueing system with redundancy. The key feature of the setting is the probability distributions of life and recovery times are assumed arbitrary. The characteristics include stationary state probabilities, state probabilities over the regeneration periods and other reliability measures (time to fist failure and time between failures distributions, regeneration period mean length). 

The paper is well written, the quality of the presentation is quite satisfactory. All the results and their derivations are mathematically sound. The numerical simulation confirms that the presented probabilistic characteristics are sensitive to the distributions of life and recovery times.

The following improvements are advisable to make the paper more reader-friendly:

More detailed model description with an explanation of the specific terms such as cold redundancy. A transition graph is desirable. A detailed caption and better explanation for figure 1, equations (5) and (16) are welcome. The latter two are claimed to follow directly from the former, but unfortunately, this deduction does not seem obvious. The abbreviations are redundant, consider at least bringing them all together in the list in the end of the paper.

Minor issues: 

Ambiguous notation for the time until the first failure, F_1, F^(1). Consider reformulation of the statement in lines 154-155, since it sounds like the unnecessity of the derivatives solely allows the Nelder-Mead method to converge to the minimum).

Author Response

We thank a reviewer for a very useful comments.

1. A definition of cold redundancy is given on Page 2.

2. A transition graph is given on Page 3.

3. We described in more detail Figure 1 and relations (5) and (16).

4. We have removed needless abbreviations especially in the titels of sections.

5. Misprints were corrected.

6.  Statement about  Nelder-Mead method was reformulated.